# A New Immersive Rehabilitation Therapy (MoveR) Improves More Than Classical Visual Training Visual Perceptual Skills in Dyslexic Children

**DOI:** 10.3390/biomedicines11010021

**Published:** 2022-12-22

**Authors:** Charlotte Gibert, Florent Roger, Emmanuel Icart, Marie Brugulat, Maria Pia Bucci

**Affiliations:** 1Clinique de l’Europe, 73 Boulevard de l’Europe, 76100 Rouen, France; 2Cabinet Orthoptie, 5 Rue Alfred Sisley, 17000 La Rochelle, France; 3Scale-1 Portal, 12 Avenue des Prés, 78180 Montigny le Bretonneux, France; 4MoDyCo, UMR 7114 CNRS, Paris Nanterre University, 92001 Nanterre, France

**Keywords:** dyslexia, children, new immersive rehabilitation therapy (MoveR), visual training, visual perceptual skills, TVPS

## Abstract

**Highlights:**

**What are the main findings?**
A new immersive rehabilitation therapy (MoveR) tool based to reinforce visual dis-crimination, visual attention, saccadic and vergence system and spatial orien-tation in dyslexics has been developed.
**What is the implication of the main finding?**
This new immersive rehabilitation therapy (MoveR) is able to improve visual per-ceptual skills better than classical visual training in children with dyslexia.

**Abstract:**

In this study, we wonder how to compare the improvement in visual perceptual skills (by using the test of visual perceptual skills, TVPS) in children with dyslexia after two visual training types (a new immersive rehabilitation therapy called MoveR, and the classical vision therapy). Thirty-nine children with dyslexia were enrolled in the study. They were split into two groups (G1 and G2) matched in IQ (intelligence quotient), sex, and age. Children of the group G1 underwent to MoveR training while children of the group G2 underwent to visual training. TVPS scores of four subtests were assessed twice before and 6 months after the two different types of training (MoveR or visual). MoveR training is an immersive therapy to reinforce visual discrimination, visual attention, saccadic/vergence system and spatial orientation. Visual therapy is based by training different types of eyes movements (horizontal, vertical and oblique pursuits and saccades, convergence and divergence movements), reading task and some exercise for improving eyes–head coordination. Each training type lasted 30 min a day, five days a week, for two weeks. Before training, the TVPS scores of the four subtests measured were statistically similar for both groups of children with dyslexia (G1 and G2). After training, both group of children (G1 and G2) improved the TVPS score of the four subtests assessed; however, such improvement reached significance in G1 only. We conclude that MoveR training could be a more useful tool than classical visual training to improve visual perceptual abilities in dyslexic children. Follow up studies on a larger number of dyslexic children will be necessary in order to explore whether such improvement persists over time and its eventual implication in reading or other classroom’s activities.

## 1. Introduction

Dyslexia is a specific deficit in the development of reading skills, despite normal intelligence and adequate schooling without any sensory or neurological difficulties [1]. This neurodevelopmental disorder affects 5 to 10% of the school-age population [2]. Several hypotheses have been advanced for the etiology of dyslexia: the first common hypothesis suggests a phonological deficit in dyslexia (see review [3]) but other studies have demonstrated that visual perception abnormalities could play an important role in dyslexia [4,5,6,7].

According to Nicolson et al. [8] motor coordination deficits in children with dyslexia could be due to cerebellar impairment; indeed, several studies [9,10,11,12] reported insufficient integration of visual and sensorial information by the cerebellum necessary for automatized motor capabilities in the dyslexic population.

Leisman [13] reported a relationship between reading difficulties and the inability to conceptualize and discriminate forms; Denckla and Rudel [14] demonstrated that dyslexic children fail in picture naming tests. Several studies also examined the relationship between reading ability, eye movements and visual perceptual processes [15,16,17,18]. All these studies suggested that dyslexics have more perceptual difficulties than normal readers in visual fixation and visual tracking. Recall that correct and efficient visual perceptual abilities are necessary for learning to read. Indeed, Ho, Chan, Tsang, and Lee [19] advanced the hypothesis that the deficits in visual perception reported in dyslexics could be at the origin of their difficulties in reading and writing skills. According to Evans [20], when one person reads, visual perception plays a major role in order to recognize the word; afterwards, the phonetic and semantic process takes place to read the word correctly. Consequently, visual perceptual deficiency may contribute to the learning difficulties of dyslexics. Note also that visual perception is essential for extracting visual information in the natural environment and for several school activities.

More recently, Provazza et al. [21] reported that dyslexic participants, in addition to phonological deficits, also had deficiencies in processing visual and visuo-spatial information. Using several visual and phonological tasks, these authors observed that both phonological and visual abilities are important in dyslexia, following the triangle model in which reading difficulties could be due to a deficit in three distinct systems (vision, phonology, and semantics) working together during reading.

The test of visual perceptual skills (TVPS) is a tool allowing one to assess visual perceptual skills without using motor abilities [22], and it is used by clinicians and therapists to evaluate visual perceptual abilities.

For instance, Mantovani et al. [23] compared visual perception skills using the TVPS–3 subtests in a group of dyslexic and non-dyslexic children, and they observed a statistically significant difference in the majority of the subtests. All these perceptual skills are important for reading, writing, and spelling, particularly because several words have to be learned using visual recognition; also, the poor performances in background figure tasks, related to an inability to perceive and locate an object in a given space, could cause difficulties in dyslexics to locate correct information within a text, affecting their levels of concentration and attention.

Research has been focused on training perceptual visual abilities in order to improve reading performance in the dyslexic Chinese population [24], and Gori and Facoetti [25] suggested that perceptual learning training could facilitate reading skills in dyslexics.

Fusco et al. [26] explored the effect of visual perception training (based on exercises of visual discrimination, visual memory, visual-spatial relationship, shape constancy, sequential memory, visual figure-ground coordination, and visual closure) in dyslexic children. Training was applied during 12 sessions, 50 min each, twice a week. These authors observed that after the intervention, visual perceptual abilities using the TVPS (TVPS-3); in particular, visual discrimination test, memory, visual–spatial relationship, form constancy, visual sequential memory, visual figure-ground, and visual closure skills, improved significantly in dyslexic children. Leung et al. [27] measured visual perceptual skills after 10-weeks of visual perceptual training (30 min per day) in a small group of dyslexic children (n = 8 subjects) and they reported improvement in visual perceptual skills, assessed using the revised version of the TVPS that persisted three and six months after training, suggesting a learning effect.

Based on these findings, in the present study we aim to evaluate the improvement in visual perception abilities in children with dyslexia using a new immersive reeducation tool (MoveR) based on reinforcing visual discrimination, visual attention, saccadic and vergence system, and spatial orientation.

To evaluate visual perception capabilities, the Test of Visual Perceptual Skills (TVPS-4) was used before and after training protocols. In the present study, four subtests of the TVPS have been assessed (visual discrimination, visual memory, visual spatial relationships, and visual form constancy).

## 2. Materials and Methods

### 2.1. Participants

Visual examination was performed for each child included in the study. All children had normal visual capabilities (ametropia was corrected by total optical correction after cycloplegia), and both eyes scoring at least 10/10. All children had a mean intelligent quotient (IQ) in the normal range (between 85 and 115 [28]).

Clinical characteristics of children are shown in Table 1. The one-way ANOVA failed to report any statistical difference between G1 and G2 for the age (F(1,37) = 0.09, *p* > 0.8) ELFE test for determining reading age (F(1,37) = 0.08, *p* > 0.7), and IQ scores (F(1,37) = 1.00, *p* > 0.8).

The investigation followed the principles of the Declaration of Helsinki and was approved by our Institutional Human Experimentation Committee (Comité de Protection des Personnes CPP). Written consents were obtained from the children’s parents after the experimental procedure was explained to them.

### 2.2. TVPS-4 Evaluation

Four subtests of the TVPS-4 were used to evaluate the visual perception abilities in dyslexic children: (i) Visual Discrimination (VD), in which the child is shown a picture or design and asked to identify the matching design at the bottom of the page; (ii) Visual Memory (VM), in which the child is shown a picture for 5 sec; afterwards, the child is asked to identify the matching picture on the new page; (iii) Spatial Relationships (SR), in which the child is shown a series of pictures and asked to identify the one that is different, they are advised that it “may differ in detail or in the rotation of all or part of the picture.” (iv) Form Constancy (FC), in which the child is asked to identify one picture on the page; it can be larger, smaller, or rotated. These four subtests were assessed twice before (Before training) and 6 months after (After Training) the MoveR training for the group G1 and the classical visual reeducation for the group G2.

### 2.3. Reeducation Training Protocol: MoveR and Visual

MoveR is a new immersive rehabilitation therapy in which the child was wearing 3D glasses and the different scenarios are controlled live by the clinician (see Figure 1); four different exercises were run: (i) Read in Motion; (ii) Battlerace; (iii) Jump in Words; (iv) Vergence movements. Each exercise will be described below.

Read in Motion (Figure 2A): In an urban environment, a child is invited to read a text, which scrolls on the road. The therapist can play on all the kinetic parameters, as well as on the representation of the displayed reading page (text, font, inclination, colors, and transparency). The reading speed as well as the accuracy is measured. This exercise reinforces visual discrimination and visual-attentional span, and promotes focal visual attention by limiting the overall perception of the word.

Battlerace (Figure 2B): A child is on a road in a science fiction environment and he/she has to catch the balls and avoid the obstacles set up by the therapist. This exercise is conducted in order to reduce the visual dependence in balance system disorders. It trains the child to improve saccades and motor coordination.

Jump in Words (Figure 2C): Child is in front of the 3D word, which gradually comes towards him. Child has to position himself in front of the loop of the letter b, d, p or q of the word, which arrives in order to pass (or even jump) in the right letter. The therapist can parameterize the words, the kinetic parameters, the environmental distractors and the expected precision in the positioning. The spatial orientation work, where the patient jumps through the letters and shifts his body in order to project it in front of the loop of the letters “bdpq”, is in order to promote the multimodal approach to spatial orientation.

Vergence movements (Figure 2D): Objects are displayed at different distances and in various shapes, and the child has to recognize them. The therapist can adjust the parameters for generating the 3D image in order to stimulate convergence and divergence movements.

The duration of the MoveR training was of 30 min per day for 5/7 days per week for 2 weeks.

Visual training protocol consisted of the reeducation of different types of eye movements: horizontal, vertical and oblique pursuits and saccades, convergence and divergence movements, reading task and some exercises to improve eye-head coordination. Similar to MoveR training, the duration of the visual training was 30 min per day for 5/7 days per week for 2 weeks.

### 2.4. Data Analysis

For each child, each correct response of each TVPS subtest was scored as “1” and a total score was recorded at the end of each subtest. Raw scores were then converted to scaled scores (0–19).

### 2.5. Statistical Analysis

ANOVA was run between the two groups (G1 and G2) of dyslexic children on the scores of each TVPS subtest recorded two times (Before and After Training). Post hoc comparisons were performed with the Bonferroni method. Significance was considered when the *p*-value was below 0.05. All statistical analyses were processed using JASP software (a free and open-source program for statistical analysis supported by the University of Amsterdam).

## 3. Results

Figure 3 summarizes the scores measured in the four different TVPS subtests in the two groups of dyslexic children (G1 and G2) recorded Before and After Training. Regarding the scores obtained in the Visual Discrimination (VD) task (see Figure 3A), the ANOVA revealed a significant training effect (F(1,37) = 48.70, *p* < 0.001 η^2^ = 0.27); the scores of the VD test after training were greater than those reported before training. There was also a significant interaction T × G effect (F(1,37) = 24.28, *p* < 0.001 η^2^ = 0.13). Post-hoc analysis revealed that after training, the score on the VD test was significantly higher in G1 than in G2 (*p* < 0.001), in the absence of any significant difference (*p* < 0.5) between G2 and G1 at baseline (Before Training). G2 pre-post was not significant different (*p* > 0.7).

Concerning the scores on Visual Memory (VM) (see Figure 3B) the ANOVA showed a significant training effect (F(1,37) = 23.82, *p* < 0.001 η^2^ = 0.16); the scores observed after training were greater than those measured before training. The ANOVA also reported a significant interaction T × G effect (F(1,37) = 11.69, *p* < 0.002 η^2^ = 0.08). The Bonferroni post-hoc test revealed that the score of the VM test in G1 increased significantly after training (*p* < 0.001), whereas before training, the scores of the VM of both groups (G1 and G2) did not reach significant difference (*p* > 0.4). The G2 pre-post was not significant different (*p* > 0.6).

The score of the Spatial Relationships (SR) test is shown in Figure 3C. The ANOVA revealed a significant training effect (F(1,37) = 78.53, *p* < 0.001 η^2^ = 0.18); the scores measured after training were greater than those measured before training. The ANOVA showed also a significant interaction T × G effect (F(1,37) = 3.84, *p* < 0.05 η^2^ = 0.02). The Bonferroni post hoc test revealed that the score of the SR test in G1 increased significantly after training (*p* < 0.001), while before training the SR scores of the two groups (G1 and G2) were similar (*p* > 0.5). The G2 pre-post was not significantly different (*p* > 0.5).

Finally, the scores obtained for the Form Constancy (FC) test are shown in Figure 3D. Similarly to the other results, the ANOVA reported a significant training effect (F(1,37) = 23.82, *p* < 0.001 η^2^ = 0.16); the scores of the FC observed after training were greater than those measured before training. The ANOVA also reported a significant interaction T × G effect (F(1,37) = 11.69, *p* < 0.002 η^2^ = 0.08). The Bonferroni post hoc test revealed that the score of the FC test in G1 increased significantly after training (*p* < 0.001);meanwhile, before training, the scores of the FC of the two groups (G1 and G2) did not reach a significant difference (*p* > 0.5). The G2 pre-post was not significantly different (*p* > 0.7).

## 4. Discussion

The purpose of the present study was to test whether a new reeducation protocol (MoveR) based on immersive training could reinforce more than classical visual training of visual and spatial discrimination, visual attention and oculomotor capabilities, to enhance visual perceptual skills in dyslexic children. Indeed, in contrast to the classical visual training, the MoveR training demonstrated significant changes in the four subtests of the TVPS assessed (visual discrimination, visual memory, visual spatial relationships, and visual form constancy). This new type of training could be used by clinicians to enhance the visual perceptual performance in the dyslexic population, as it appears to be more efficient than the classical visual training that has been used for several years. The major difference between these two types of training is that the MoveR training is an immersive therapy that intensely stimulates the visual system and allows the child to be trained from a close distance to infinity. More importantly, the MoveR training also allows one to work on different sensory inputs simultaneously (visual; vestibular) in order to develop multimodal approaches. Finally, the participants are more involved and motivated to train. All these characteristics are not present in classical visual training, given that the child is asked to be seated on a chair and the different types of eye movements are frequently trained separately. In line with frequent studies in which virtual reality has been used for training protocols in different fields (see recent work [29,30,31]), we could assume that the MoveR training could enhance visual and spatial capabilities of dyslexic children through a better attentional focus. It is well known that dyslexic participants have difficulties focusing their attention [21], and the cortical structures responsible of visual and spatial capabilities and attention skills are imbricated and overlapping [32]; consequently, we suggest that MoveR training could reinforce the interaction of all these aptitudes, leading to a better activity of this neural network.

One might question where such improvement takes place in the brain. Several studies have advanced the hypothesis that the cerebellum is important not only for the sensorimotor and vestibular control but also for cognitive functions (see review [33]). For instance, Gottwald et al. [34] reported impairment in cognitive skills and attention capabilities in patients with focal cerebellar lesions and recently, poor visual and spatial attention performance has been reported in patients with lesions in the Crus II of the left posterior cerebellum [35]. Note that a deficit in the cerebellum in dyslexic participants has been suggested for several years [36] and several subsequent studies have been reported, confirming that cerebellar abnormalities in dyslexic children (see the review [37]). Interestingly, the cerebellum is also known to be undergoing adaptive mechanisms [38]. Based on all of this knowledge, we could advance the hypothesis that the improvements reported after MoveR training could be taking place in the cerebellum, most likely leading to a better weighting of different sensorial inputs in contrast to the classical visual training that reinforces only the visual input.

Someone could advance the hypothesis that the two groups of participants could differ in terms of other cognitive abilities that were not carefully explored, given that only the IQ and reading abilities were compared in G1 and G2. Of course, a deeper comparison of cognitive skills of participants (e.g., non-serial RAN abilities, non-verbal IQ, any test such as the code test, targeting visual attention abilities) could give us more insights of the mechanism underlaying the benefit of the two programs.

Further imaging studies on a larger number of children are needed to better understand neurophysiological activities after MoveR and other training protocols. Moreover, it could be interesting to test the benefits of the two programs on other cognitive abilities.

Finally, we have to point out that MoveR training, such as visual training, can be easily used by clinicians. Perhaps clinicians, in order to take better care of children with dyslexia, might have different types of training protocols tailored to the particular and individual needs of each child.

## 5. Conclusions

The findings of the present study demonstrated that the new reeducation program (MoveR) could be a useful tool more than classical visual training for dyslexic children for improving their visual perceptual skills. Further follow up studies are needed in order to explore whether such improvement persists over time on a larger number of participants; it will also be interesting to test reading and other cognitive abilities in dyslexics after such training to better evaluate the relationship between reading and visual perceptual skills.

## Figures and Tables

**Figure 1 biomedicines-11-00021-f001:**
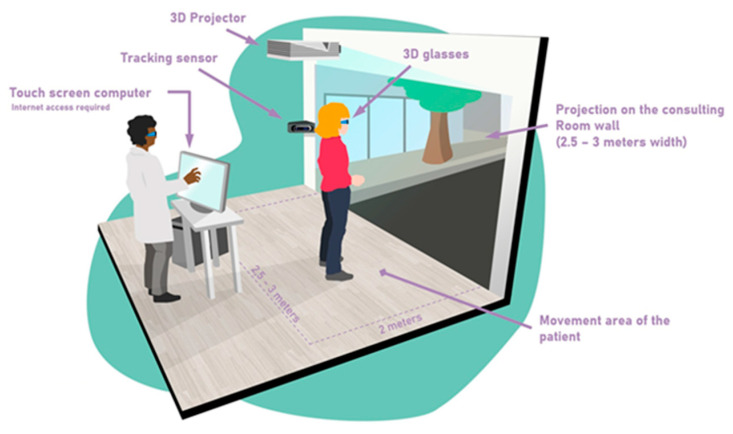
Experimental setup of MoveR training.

**Figure 2 biomedicines-11-00021-f002:**
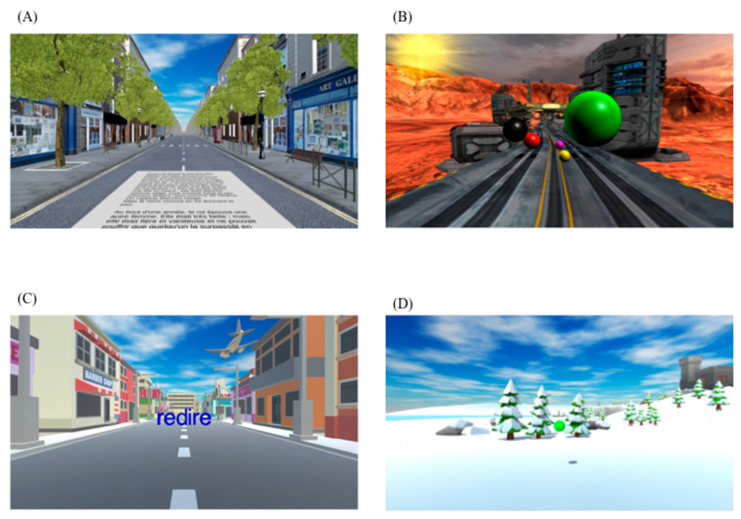
Example of MoveR exercises elicited: (**A**) Read in motion; (**B**) Battle race; (**C**) Jamp in words; (**D**) Vergence movements.

**Figure 3 biomedicines-11-00021-f003:**
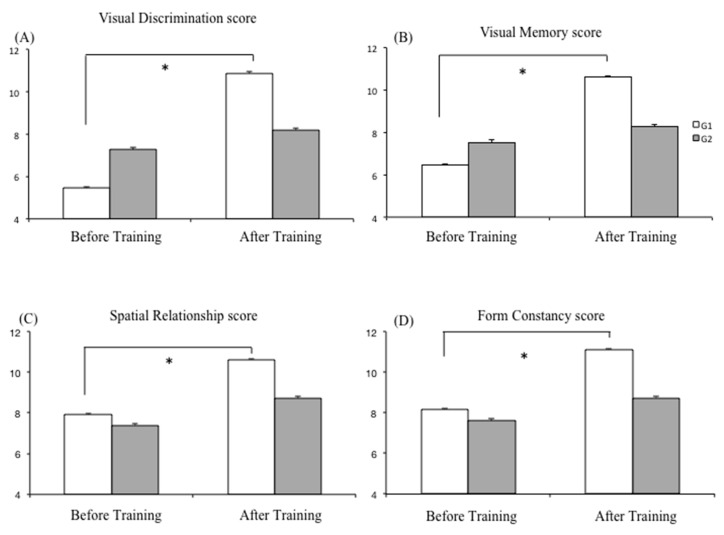
Means and standard deviations of scores measured in the different subtests of the TVPS: (**A**) Visual Discrimination, (**B**) Visual Memory, (**C**) Spatial Relationships, and (**D**) Form Constancy, for the two groups of children with dyslexia (G1 and G2) recorded Before and After Training. Asterisks indicate that the value is significantly different (*p* ≤ 0.05).

**Table 1 biomedicines-11-00021-t001:** Clinical characteristics of the two groups (G1 and G2) of children with dyslexia enrolled in the study, with the number of girls and boys, the mean and the standard deviations of the Chronological Age, the Reading Age (E.L.FE Test), and the IQ Score.

	G1 N = 24	G2 N = 15
Girls/boys	8/16	5/10
Chronological age (years)	9.0 ± 0.1	9.1 ± 0.1
Reading age (years)	7.4 ± 0.4	7.6 ± 0.6
IQ (WISC-IV)	95.9 ± 10	96.4 ± 12

## Data Availability

The datasets analyzed during the current study are available from the corresponding author on reasonable request.

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
