# Peer review of "A New Immersive Rehabilitation Therapy (MoveR) Improves More Than Classical Visual Training Visual Perceptual Skills in Dyslexic Children"

_biomedicines, 2022, doi:10.3390/biomedicines11010021_

Round 1

Reviewer 1 Report

Title, highlights and abstract are not clear in only reason – reader cannot understand what is “MoveR”

Moreover, this a new tool (MoveR) was not described based on evidence in task statement area. 

Could you introduce what does it mean MoveR in first words of article?

Could you state the problem of connection between movement and dislexia in introduction section?

Author Response

Title, highlights and abstract are not clear in only reason – reader cannot understand what is “MoveR”
According to your suggestion we better explained what is MoveR (new immersive rehabilitation therapy.
Moreover, this a new tool (MoveR) was not described based on evidence in task statement area. 
The new immersive rehabilitation therapy (MoveR) is explained in detail at pages 4, 5 and in Figures 1 and 2 is shown the set up and the different exercises.
Could you introduce what does it mean MoveR in first words of article?
We agree with you, from the beginning we added ‘new immersive rehabilitation therapy (MoveR).
Could you state the problem of connection between movement and dyslexia in introduction section?
Following your suggestions we added the cerebellar deficit hypothesis.

Reviewer 2 Report

The authors tested two visual perception training programs – MoveR and a classic program. – in two groups of dyslexic children matched for reading age. They found an advantage of MoveR over the classic training on visual-perceptual abilities.

The article is clearly written and well-organized. I have, however, some concerns regarding the consistency between the paradigm and the conclusions drawn (maybe the most critical issue), the sample, the null results and the discussion.

Paradigm vs. conclusions

The paradigm compares the effects of two different methods, but it does not include a control group (no program or, ideally, program targeting something other than visual perception). In this sense, I would say that there are no reasons to say that MoveR “improved” partcipants’ skills – we do not know how performance would evolve if they had had no training or perception-unrelared training (theoretically, they could improve more than they did in G1 and, in this case, there would be a disadvantage of MoveR). I would say that what the study tests is: which program is better?

Participants

-usually the word subjects is replaced by participants to avoid the negative meaning associated to the former

-Was there any power analysis?

- Only IQ and reading age are reported as sample characteristics. Could it be that any other differences between G1 and G2 could account for the different benefits of the two programs (e.g., non-serial RAN abilities, non-verbal IQ, any test like code test, targeting visual attention abilities)?This could be debated in the discussion.

Results

-Please report the statistics for null results regarding baseline comparisons G1-G2 and G2 pre-post.

Discussion

-Please clarify how the current results (i.e., the fact that MoveR and not the classic program worked) speak in favor of a cerebellar deficit

Typos

Ln 240 – where in the brain (not whether) such improvement takes place ?

Ln 260-could take place/could be taking place

Author Response

The authors tested two visual perception training programs – MoveR and a classic program. – in two groups of dyslexic children matched for reading age. They found an advantage of MoveR over the classic training on visual-perceptual abilities.
The article is clearly written and well-organized. I have, however, some concerns regarding the consistency between the paradigm and the conclusions drawn (maybe the most critical issue), the sample, the null results and the discussion.
Paradigm vs. conclusions
The paradigm compares the effects of two different methods, but it does not include a control group (no program or, ideally, program targeting something other than visual perception). In this sense, I would say that there are no reasons to say that MoveR “improved” partcipants’ skills – we do not know how performance would evolve if they had had no training or perception-unrelared training (theoretically, they could improve more than they did in G1 and, in this case, there would be a disadvantage of MoveR). I would say that what the study tests is: which program is better?
We agree with you. The title, highlights, discussion and conclusion changed following your suggestion
Participants
-usually the word subjects is replaced by participants to avoid the negative meaning associated to the former
Complied
-Was there any power analysis?
Actually we did not conduct a power analysis for this study. This point was added in the conclusion for further studies on a larger number of participants
- Only IQ and reading age are reported as sample characteristics. Could it be that any other differences between G1 and G2 could account for the different benefits of the two programs (e.g., non-serial RAN abilities, non-verbal IQ, any test like code test, targeting visual attention abilities)?This could be debated in the discussion.
We agree with you; this important point has been added.
Results
-Please report the statistics for null results regarding baseline comparisons G1-G2 and G2 pre-post.
Null results were added.

Discussion
-Please clarify how the current results (i.e., the fact that MoveR and not the classic program worked) speak in favor of a cerebellar deficit
See Discussion section page 8.
Typos
Ln 240 – where in the brain (not whether) such improvement takes place ?
Complied
Ln 260-could take place/could be taking place
Complied

Round 2

Reviewer 2 Report

The authors have addressed most of my comments, but I still have one concern.

- Only IQ and reading age are reported as sample characteristics. Could it be that any other differences between G1 and G2 could account for the different benefits of the two programs (e.g., non-serial RAN abilities, non-verbal IQ, any test like code test, targeting visual attention abilities)?This could be debated in the discussion.
We agree with you; this important point has been added.

I do not think that the authors have discussed the point that I raised (differences across groups could have been due to other characteristics of the two groups). Instead, they mentioned the possibility of targeting improvements in Ran and the other skills I mentioned in future studies. These are different things.

Author Response

Dear reviewer, we corrected the Discussion following your suggestion